# Factors associated with patient experiences of the burden of using medicines and health-related quality of life: A cross-sectional study

**Won Sun Chen**[1]*, **Md. Rafiqul Islam**[2,3], **Sajini Ambepitiya**[2], **William Sim**[2], **Wai Yiu**[2], **Joseph Carey**[2], **Edward Ogden**[1,2]

**1** School of Health Sciences, Swinburne University of Technology, Hawthorn, Victoria, Australia, **2** Goulburn Valley Health, Shepparton, Victoria, Australia, **3** School of Rural Health, La Trobe University, Shepparton, Victoria, Australia

\* wchen@swin.edu.au

## Abstract

### Objective

Polypharmacy, defined as the concurrent use of multiple medications, is a growing concern globally. This study aimed to identify the significant factors that predict the perceived burden of medication and health-related quality of life.

### Methods

Adults, aged 18 years and above who have used at least two regular medicines, were invited to complete the study questionnaires between June and October 2019. Multiple linear regression analysis was conducted to identify significant predictors for perceived burden of medication and health-related quality of life.

### Results

A total of 119 participants completed this study. The average age of the participants was 63 years (SD±16 years). Factors significantly predicting perceived burden of medication were participants' current health condition (p = 0.001), overall burden of treatment (p<0.001) and being hypertensive (p = 0.037). Similarly, participants' current health condition (p<0.001) and overall burden of treatment (p = 0.086) were significant predictors for perceived health-related quality of life.

### Conclusions

This study revealed that hypertensive participants in poor health tended to experience higher perceived burden of medication, which in turn was found to be correlated with lower perceived health-related quality of life.

**Data Availability Statement:** The authors have deposited their data to Dryad Digital Repository. The dataset has been assigned a unique digital

object identifier (DOI): doi:10.5061/dryad.
d2547d84m.

**Funding:** The authors received no specific funding
for this work.

**Competing interests:** The authors have declared
that no competing interests exist.

## Introduction

Polypharmacy, defined as the concurrent used of several medicines, is a growing concern globally [1–3]. The trend of increased prescribing of drugs for secondary prevention, poses an increasing burden to some patients [4–7]. Specifically, almost a third of patients over the age of 60 years use five or more medicines frequently, and polypharmacy has been found to be negatively associated with socioeconomic status [8]. Additionally, polypharmacy has also been found to be associated with adverse outcomes, such as increased hospitalisation, cognitive impairment, falls, and drug interactions [8]. Polypharmacy has been posited in past studies as one of the reasons patients are reluctant to take medicines [9, 10].

In Australia, the prevalence of polypharmacy was estimated to be between 43% and 91% [11–14]. The higher polypharmacy was found to be associated to those individuals with greater needs, including hospital inpatients and aged care residents. The disparity of these estimates is partially due to a lack of a uniform approach in defining polypharmacy. For example, exposure to medicines can be estimated according to dispensing claims, prescriptions, or patient self-reported numbers [15].

Several instruments exist to measure satisfaction with medicines [16, 17] and the overall impact of using medicines on the quality of life [18]. The long-term use of medicine is, however, multidimensional and complex. Any individual can experience both positive and negative aspects of medicine use [6, 7, 19, 20]. Medicine-related burden is a relatively new concept. It includes impacts on behaviours (such as non-adherence); practical difficulties (such as opening packaging); challenges with managing complex regimes; psychosocial issues, particularly social stigma; disruptions to daily living; and health system burden associated with regular medicine use [20–24].

The Living with Medicines Questionnaire (LMQ) was developed primarily to measure perceived burden of medication [23]. This instrument consists of 60 items, accompanied by a five-point Likert scale (strongly agree to strongly disagree) and a free-text open question. The LMQ has been demonstrated to be a valid and reliable multidimensional measure of prescription medicine use experiences. Side effects are further strengthened within the LMQ version 3 (LMQ-3) into a separate domain and proved to be one of the questionnaires that most strongly associated with perceived burden of medication [25, 26].

In this study, we primarily examined the relationship between perceived burden of medication, overall burden of treatment and health-related quality of life with demographic factors. The secondary objective was to identify significant factors that predict perceived burden of medication and health-related quality of life.

## Materials and methods

### Study design and participants

This cross-sectional study was conducted in Goulburn Valley Health, Victoria, Australia, between the period of June and October 2019 among people aged 18 years and above who used at least two regular medicines. A study advertisement was posted on the notice boards located throughout Goulburn Valley Health, and interested participants were asked to contact the study researchers through the contact details provided on the advertisement. After obtaining written informed consent, all participants were invited to complete a set of the following questionnaires available through a paper-based method or an online Qualtrics platform. This study was approved by Swinburne University Human Research Ethics Committee (SHR Project 2019/108) and Goulburn Valley Health Human Research Ethics Committee (LNR/51946/GVH-2019-169364(v2)).

### Research questionnaire

The research questionnaire consisted of demographic questions and the following well-validated scales: LMQ-3 and EQ-5D-5L questionnaire. Permission to use the LMQ-3 (UK English version) and EQ-3D (UK English version) was granted by the Universities of Greenwich and Kent at Medway as well as the EuroQol Research Foundation at Rotterdam, respectively.

### Living with Medicines Questionnaire version 3 (LMQ-3)

The LMQ-3 consists of 41 items, measured using a five-point Likert scale (scored from strongly agree to strongly disagree) within eight domains: 1) perceptions about effectiveness, 2) concerns about medicine use, 3) patient-provider relationships and communication about medicines, 4) practical difficulties, 5) interferences with daily life, 6) side effects, 7) costs, 8) autonomy/control over medicine and acceptance of medicine use, all of which have been cited by users of long-term medicines as burdensome [20, 26].

Domain scores are tallied to produce a total score (total LMQ-3 score) representing the perceived burden of medication. The total score ranges from 41 to 205, with higher scores reflecting higher perceived burden of medication. The total score can be further categorised into 1) no/minimal burden (41–87), 2) moderate degree of burden (88–110) and 3) high burden, potentially benefitting from intervention (111–205). In addition, a 10 cm visual analogue scale (VAS) ranging from 0 (no burden at all) to 10 (extremely burdensome) enables self-reflection of overall burden of treatment (VAS-burden). This VAS-burden score can be further categorised into 1) no/minimal burden (0–4), some degree of burden (4.1–5.9) and 3) high degree of burden (6–10) [23].

### EQ-5D-5L questionnaire

The EQ-5D-5L is a standardised measure of health-related quality of life. It consists of 2 pages: the EQ-5D-5L descriptive system and the EQ visual analogue scale (EQ VAS). The descriptive system comprises 5 dimensions: mobility, self-care, usual activities, pain/discomfort, and anxiety/depression. Each dimension has 5 levels: 1 = no problems, 2 = slight problems, 3 = moderate problems, 4 = severe problems, and 5 = extreme problems. The EQ VAS captures one's self-reported overall health on a 20 cm vertical, visual analogue scale with endpoints of 0 (the worst health you can imagine) and 100 (the best health you can imagine) [27].

### Statistical analysis

Continuous data was summarised using descriptive statistics (such as mean, standard deviation, median and range), while frequency and percentage were presented for categorical data.

Correlations between perceived burden of medication, overall burden of treatment, health-related quality of life, and demographic factors were assessed using Spearman correlation coefficients. Relationships between demographic characteristics and perceived burden of medication as well as health-related quality of life were explored using parametric tests such as independent samples t-tests and Analysis of Variance (ANOVA). Due to significant departure from normality, nonparametric tests such as Mann-Whitney U test and Kruskal-Wallis test were performed to assess the relationship with overall burden of treatment.

Multiple linear regression analysis was performed to examine factors affecting the perceived burden of medication and perceived health-related quality of life respectively, after adjusted for significant demographic factors. Diagnostic testing was conducted for outliers (deleted standardised residuals), influential points (Mahalanobis Distance), linearity (partial correlation plots) and multicollinearity (variance inflation factors). A p-value $< 0.05$ was deemed

statistically significant for all 2-sided tests. The analyses were conducted using IBM SPSS Statistics version 27 (IBM Corp., Armonk, NY, USA).

## Results

### Descriptive statistics

A total of 119 participants were recruited for this study. Table 1 shows a summary of the demographic data. Participants were aged between 18 and 95 years with an average age of 63 years (SD±16 years), with a predominance of retired female Australian participants with primary/secondary educational attainment (61%) and at least moderate health condition (90%). The top three medical conditions reported were arthritis (63%), high blood pressure (57%) and heart problems (33%).

At least half of the participants reported taking 5 medicines concurrently, in tablet and capsule formulations (98%), once daily (54%). Almost all participants managed and paid for their own medicines (Table 1).

### Correlation between LMQ-3 scores and EQ VAS scores

Increasing age was found to be correlated with higher number of medical conditions ($r = 0.481$, $p < 0.010$), and also higher number of medicines used ($r = 0.376$, $p < 0.010$). Additionally, participants in the older age group tended to report experiencing lower perceived burden of medication ($r = -0.161$, $p = 0.081$) and lower overall burden of treatment ($r = -0.249$, $p = 0.006$), which resulted in negative impacts on health-related quality of life ($r = -0.027$, $p = 0.773$). Poor health was found to be significantly correlated with higher perceived burden of medication ($r = 0.436$, $p < 0.010$), higher overall burden of treatment ($r = 0.297$, $p < 0.010$), and lower health-related quality of life ($r = -0.635$, $p < 0.010$). On the other hand, a significant positive correlation was found between the number of medicines used and the number of medical conditions ($r = 0.562$, $p < 0.010$). Furthermore, there was a negative relationship between health-related quality of life and the number of medicines used ($r = -0.326$, $p < 0.010$) as well as the number of medical conditions ($-0.112$, $p = 0.223$).

### Relationship between LMQ-3 scores, VAS-burden scores and EQ VAS scores with demographics

Table 2 shows that the perceived burden of medication was significantly different across varying health conditions ($F_{(3,115)} = 8.60$, $p < 0.001$), particularly for hypertensive participants ($F_{(1, 117)} = 9.03$, $0 = 0.003$). Overall burden of treatment was found to be significantly different across age groups ($H_{(6)} = 15.27$, $p = 0.018$) and self-rated health condition ($H_{(3)} = 13.29$, $p = 0.004$). Interestingly, perceived health-related quality of life was significantly different across highest educational attainment ($F_{(2, 116)} = 3.22$, $p = 0.044$) and self-rated health condition ($F_{(3, 115)} = 29.25$, $p < 0.001$).

A further analysis of the LMQ-3 domains indicates that the average score for interferences with social or leisure activities and daily tasks domain was found to be significantly different across the age groups ($F_{(6, 112)} = 3.07$, $p = 0.008$). The average domain score for the relationships with healthcare professionals relating to medicines ($F_{(3, 115)} = 2.84$, $p = 0.041$), practical difficulties in getting prescriptions from the doctor ($F_{(3, 115)} = 4.35$, $p = 0.006$), lack of perceived effectiveness of their medicines ($F_{(3, 115)} = 3.73$, $p = 0.013$), experience of bothersome side effects ($F_{(3, 115)} = 6.03$, $p < 0.001$), general concerns about long-term effects of using medicines ($F_{(3, 115)} = 6.49$, $p < 0.001$), cost-related burden ($F_{(3, 115)} = 5.19$, $p = 0.002$), as well as interferences with daily life ($F_{(3, 115)} = 8.70$, $p < 0.001$) were found to be significantly different

**Table 1. Demographic and medicine-related characteristics (n = 119).**

| Demographic characteristics | Total (n = 119) |
|---|---|
| Age (years) | |
| Median (Range) | 65 (18–95) |
| Mean ± SD | 63.4 ± 15.5 |
| Gender, n (%) | |
| Female | 67 (56.3%) |
| Male | 52 (43.7%) |
| Nationality, n (%) | |
| Australian | 108 (90.8%) |
| Non-Australian | 11 (9.2%) |
| Highest Educational Attainment, n (%) | |
| No formal education | 1 (0.8%) |
| Primary / Secondary | 72 (60.5%) |
| Tertiary | 46 (38.7%) |
| Employment Status, n (%) | |
| Employed | 22 (18.5%) |
| Unemployed | 13 (10.9%) |
| Retired | 65 (54.6%) |
| Full-time / part-time student | 7 (5.9%) |
| Others | 12 (10.1%) |
| Self-Rated Health Condition | |
| Excellent | 3 (2.5%) |
| Good | 46 (38.7%) |
| Moderate | 58 (48.7%) |
| Poor | 12 (10.1%) |
| History of Medical Condition(s) | |
| High blood pressure | 68 (57.1%) |
| Diabetes | 30 (25.5%) |
| Heart problems | 39 (32.8%) |
| Stroke | 18 (15.1%) |
| Cataract / glaucoma | 23 (19.3%) |
| Hearing problem | 25 (21.0%) |
| Arthritis | 75 (63.0%) |
| Dementia | 3 (2.5%) |
| **Medicine-related characteristics** | |
| No. of medicines | |
| Median (Range) | 5 (1–25) |
| Mean ± SD | 6 ± 3.9 |
| Formulation used | |
| Tablets/capsules | 117 (98.3%) |
| Other formulations | 17 (14.3%) |
| Both types | 16 (13.4%) |
| Frequency of use | |
| Once per day | 64 (53.8%) |
| Twice per day | 54 (45.4%) |
| Three times per day | 19 (16.0%) |
| More than 3 times per day | 7 (5.9%) |
| Other times | 5 (4.2%) |

(*Continued*)

**Table 1.** (Continued)

| Demographic characteristics | Total (n = 119) |
|---|---|
| Managing medicines | |
| Yes (Required assistance) | 23 (19.3%) |
| Paying for prescription | |
| Yes | 113 (95.0%) |

across varying health conditions. Hypertensive participants generally reported lower perceived burden of medication across all eight domains. Overall, the Cronbach's alpha for all domains were found to be acceptable (ranged from 0.646 to 0.821), except for practical difficulties in getting prescriptions from the doctor (Cronbach's alpha = 0.461) (Table 3).

Table 3 shows the median score of health-related quality of life was significantly different across varying health conditions (F(3, 115) = 29.25, p<0.001). This finding was consistent across all four domains in EQ-5D-5L; difficulties in mobility (H(3) = 21.22, p<0.001); self-care (H(3) = 13.84, p = 0.003); usual activities (H(3) = 20.55, p<0.001); and pain or discomfort (H(3) = 25.44, p<0.001), except for anxiety or depression (H(3) = 4.04, p = 0.257). The median score for difficulties in mobility domain was found to be significantly different across the age groups (H(6) = 13.46, p = 0.036).

## Regression analysis

Increasing age was found to be correlated with higher number of medical conditions (r = 0.481, p<0.010), and also higher number of medicines used (r = 0.376, p<0.010). Therefore, it was important to adjust for these variables in the regression analysis.

After controlling for age, number of medicines used and number of medical conditions, the regression analysis revealed that being hypertensive (β = -7.52, p = 0.037), self-rated health condition (β = 8.84, p = 0.001) and overall burden of treatment (β = 3.37, p<0.001) were significant in predicting the perceived burden of medication. On the other hand, only self-rated health condition (β = -18.77, p<0.001) and overall burden of treatment (β = -2.16, p = 0.086) were significantly predicting health-related quality of life, after adjusted for age, number of medicines used and number of medical conditions (Table 4). All relevant diagnostics for outliers, influential points, linearity, and multicollinearity were performed and verified.

## Post-hoc calculation of sample size

Using G*Power (version 3.1.9.4) post-hoc calculation with a medium effect size, a 5% significance level, six predictors and a sample size of 119 resulting in the power of the study to be 89%.

## Discussion

Age was found to be positively correlated with both number of medical conditions and number of medicines used. Interestingly, older participants tended to experience a lower perceived burden of medication and overall burden of treatment, as well as all domain scores except for the autonomy domain. These findings are consistent with that of other studies [21, 25]. It is possible that older people have been taking medicines regularly for a longer period in comparison to younger people. Therefore, they may have developed regular routines for managing the required medicines [25].

**Table 2. Demographic and medicines used characteristics on LMQ-3 total score, VAS-burden score and EQ VAS score.**

| Characteristics | Perceived Burden of Medication | | Overall Burden of Treatment | | Perceived Health-Related Quality of Life | |
|---|---|---|---|---|---|---|
| | (LMQ-3 Total Score) | | (VAS-burden Score) | | (EQ VAS Score) | |
| | Mean score (SD) | p-value | Median score (min-max) | p-value | Mean score (SD) | p-value |
| Age (years) | | | | | | |
| 18–29 | 131.0 (17.0) | 0.067[a] | 4.5 (4–5) | 0.018[b] | 62.50 (31.8) | 0.955[a] |
| 30–39 | 109.3 (22.7) | | 3.0 (2–6) | | 55.71 (29.9) | |
| 40–49 | 100.6 (16.0) | | 8.5 (0–9) | | 61.3 (19.0) | |
| 50–59 | 110.6 (20.4) | | 5.0 (0–10) | | 59.9 (19.3) | |
| 60–69 | 97.6 (20.1) | | 1.0 (0–10) | | 63.0 (21.8) | |
| 70–79 | 96.5 (19.0) | | 1.0 (0–7) | | 61.8 (22.4) | |
| ≥80 | 99.7 (22.6) | | 1.0 (0–7) | | 56.6 (16.9) | |
| Gender | | | | | | |
| Female | 99.8 (22.4) | 0.146[a] | 2.0 (0–10) | 0.255[b] | 59.4 (21.2) | 0.424[a] |
| Male | 104.4 (17.4) | | 3.0 (0–10) | | 62.5 (20.5) | |
| Highest Educational Attainment, n (%) | | | | | | |
| Primary / Secondary | 99.1 (20.2) | 0.159[a] | 1.0 (0–8) | 0.054[b] | 61.9 (21.5) | 0.044[a] |
| Tertiary | 104.5 (21.0) | | 3.0 (0–10) | | 60.0 (18.7) | |
| Employment Status, n (%) | | | | | | |
| Employed | 99.3 (18.7) | 0.076[a] | 3.0 (0–9) | 0.059[b] | 66.0 (18.3) | 0.534[a] |
| Unemployed | 114.2 (22.4) | | 4.0 (0–10) | | 58.8 (20.5) | |
| Retired | 97.9 (19.8) | | 1.0 (0–10) | | 59.1 (21.3) | |
| Full-time / part-time student | 109.6 (23.9) | | 3.0 (1–8) | | 54.4 (28.7) | |
| Others | 103.8 (19.8) | | 1.0 (0–8) | | 65.6 (18.6) | |
| Self-Rated Health Condition | | | | | | |
| Excellent | 91.0 (26.2) | <0.001[a] | 0.0 (0–1) | 0.004[b] | 92.7 (6.8) | <0.001[a] |
| Good | 90.9 (16.0) | | 1.0 (0–6) | | 73.2 (13.8) | |
| Moderate | 107.9 (21.0) | | 3.5 (0–10) | | 55.1 (17.6) | |
| Poor | 111.5 (15.3) | | 3.0 (0–7) | | 31.7 (16.1) | |
| History of Medical Condition(s) | | | | | | |
| High blood pressure | 96.5 (18.3) | 0.003[a] | 2.0 (0–9) | 0.183[b] | 59.9 (21.7) | 0.624[a] |
| Diabetes | 104.5 (18.9) | 0.311[a] | 3.0 (0–10) | 0.037[b] | 58.1 (21.0) | 0.424[a] |
| Heart problems | 100.0 (21.4) | 0.655[a] | 2.5 (0–10) | 0.363[b] | 59.4 (20.4) | 0.644[a] |
| Stroke | 100.3 (17.9) | 0.841[a] | 2.5 (0–8) | 0.958[b] | 52.3 (21.9) | 0.064[a] |
| Cataract / glaucoma | 98.4 (18.5) | 0.462[a] | 1.0 (0–7) | 0.341[b] | 55.9 (20.9) | 0.216[a] |
| Hearing problem | 98.9 (20.7) | 0.521[a] | 1.0 (0–8) | 0.145[b] | 62.2 (19.9) | 0.690[a] |
| Arthritis | 100.4 (20.2) | 0.552[a] | 2.0 (0–10) | 0.707[b] | 60.5 (21.4) | 0.890[a] |
| Dementia | 109.0 (27.1) | 0.509[a] | 6.5 (2–8) | 0.113[b] | 51.7 (20.2) | 0.449[a] |

[a] Independent samples t-test/One-Way Analysis of Variance;

[b] Mann-Whitney U test/Kruskal-Wallis test

In addition, a negative correlation was revealed between age and health-related quality of life. The findings show that health-related quality of life was lower for those aged 70 years and above. Consequently, it is likely that older people perceive medicine as a "necessity" rather than a "burden", which could ultimately be correlated with a lower health-related quality of life [25].

Additionally, poor self-rated health condition was found to be correlated with higher levels of perceived burden of medication and overall burden of treatment, and lower health-related

**Table 3. LMQ-3 and EQ-5D-5L domain scores with significant demographic factors.**

| Factor | LMQ-3 Domain (maximum score) | | | | | | | | |
|---|---|---|---|---|---|---|---|---|---|
| | Mean Domain Score (SD) | | | | | | | | |
| | Relationships (25) | Practicalities (35) | Lack of Effectiveness (30) | Side Effects (20) | Concerns (35) | Cost (15) | Interferences (30) | Autonomy (15) | Perceived Medicine Burden (201) |
| Cronbach alpha | 0.746 | 0.461 | 0.786 | 0.792 | 0.821 | 0.839 | 0.646 | 0.660 | 0.901 |
| Age (years) | | | | | | | | | |
| 18–29 | 12.5 (3.5) | 22.0 (2.8) | 14.0 (7.1) | 12.5 (3.5) | 26.0 (1.4) | 12.0 (1.4) | 19.5 (0.7) | 12.5 (2.1) | 131.0 (17.0) |
| 30–39 | 10.9 (2.6) | 17.6 (2.5) | 13.1 (3.3) | 10.1 (2.0) | 21.1 (6.8) | 8.4 (4.2) | 18.3 (4.1) | 9.7 (2.8) | 109.3 (22.7) |
| 40–49 | 10.5 (2.7) | 16.0 (3.7) | 13.4 (3.6) | 9.6 (2.9) | 18.3 (5.1) | 7.6 (3.4) | 15.6 (3.6) | 9.7 (2.0) | 100.6 (16.0) |
| 50–59 | 10.4 (3.4) | 18.0 (3.1) | 13.7 (3.5) | 10.0 (3.5) | 21.9 (6.0) | 8.7 (3.2) | 16.9 (5.5) | 10.8 (2.7) | 110.6 (20.4) |
| 60–69 | 10.1 (3.8) | 16.1 (3.7) | 11.6 (3.6) | 8.4 (3.4) | 19.7 (6.1) | 6.7 (3.2) | 13.6 (3.6) | 11.3 (3.1) | 97.6 (20.1) |
| 70–79 | 9.7 (2.7) | 16.9 (4.0) | 11.7 (3.4) | 7.6 (3.0) | 18.2 (6.5) | 6.4 (3.3) | 14.1 (3.2) | 11.8 (2.7) | 96.5 (19.0) |
| ≥80 | 10.7 (4.8) | 17.8 (4.3) | 12.6 (4.3) | 8.9 (4.4) | 18.0 (5.9) | 6.4 (3.3) | 14.8 (3.4) | 11.6 (2.6) | 99.7 (22.6) |
| p-value[a] | 0.919 | 0.232 | 0.379 | 0.138 | 0.212 | 0.063 | 0.008 | 0.219 | 0.067 |
| Self-Rated Health Condition | | | | | | | | | |
| Excellent | 12.7 (8.0) | 14.7 (3.2) | 12.0 (7.2) | 8.7 (4.2) | 14.7 (8.0) | 4.3 (1.5) | 11.1 (1.5) | 12.7 (2.1) | 91.0 (26.2) |
| Good | 9.4 (2.9) | 15.5 (2.6) | 11.1 (2.9) | 7.3 (2.7) | 16.9 (5.4) | 6.0 (2.9) | 13.1 (3.0) | 11.6 (2.5) | 90.9 (16.0) |
| Moderate | 10.6 (3.2) | 17.6 (4.0) | 13.1 (3.8) | 9.8 (3.5) | 21.3 (6.2) | 8.3 (3.5) | 16.5 (4.2) | 10.7 (3.0) | 107.9 (21.0) |
| Poor | 12.0 (4.5) | 18.5 (4.7) | 14.2 (3.9) | 10.3 (3.6) | 22.1 (4.0) | 7.7 (3.1) | 16.1 (4.0) | 10.7 (2.6) | 111.5 (15.3) |
| p-value[a] | 0.041 | 0.006 | 0.013 | 0.001 | <0.001 | 0.002 | <0.001 | 0.284 | <0.001 |
| History of Medical Condition(s) | | | | | | | | | |
| Hypertensive | 9.6 (3.2) | 16.2 (3.5) | 11.6 (3.2) | 8.0 (3.1)) | 18.9 (6.0) | 6.9 (3.3) | 14.2 (3.7) | 11.1 (3.0) | 96.5 (18.3) |
| Non-Hypertensive | 11.2 (3.6) | 17.6 (3.9) | 13.6 (4.0) | 10.0 (3.7) | 20.3 (6.3) | 7.6 (3.4) | 16.1 (4.3) | 11.2 (2.5) | 107.6 (21.7) |
| p-value[a] | 0.014 | 0.041 | 0.003 | 0.002 | 0.242 | 0.255 | 0.009 | 0.814 | 0.003 |

| Factor | EQ-5D-5L Domain | | | | | |
|---|---|---|---|---|---|---|
| | Median Domain Score (min-max) | | | | | |
| | Mobility | Self-Care | Usual Activities | Pain / Discomfort | Anxiety / Depression | Perceived Health-Related Quality of Life |
| Age (years) | | | | | | |
| 18–29 | 0.5 (1–2) | 1.0 (1–1) | 2.5 (2–3) | 3.0 (3–3) | 3.0 (2–4) | 62.50 (31.8) |
| 30–39 | 1.0 (1–2) | 1.0 (1–1) | 1.0 (1–5) | 1.0 (1–3) | 2.0 (1–4) | 55.71 (29.9) |
| 40–49 | 1.0 (1–4) | 1.0 (1–1) | 1.0 (104) | 2.0 (1–4) | 2.0 (1–3) | 61.3 (19.0) |
| 50–59 | 2.0 (1–5) | 1.0 (1–4) | 2.0 (1–5) | 2.0 (1–5) | 2.0 (1–5) | 59.9 (19.3) |
| 60–69 | 2.0 (1–4) | 1.0 (1–4) | 1.5 (1–5) | 2.0 (1–4) | 1.0 (1–3) | 63.0 (21.8) |
| 70–79 | 2.0 (1–4) | 1.0 (1–3) | 1.0 (1–5) | 2.0 (1–4) | 1.0 (1–3) | 61.8 (22.4) |
| ≥80 | 2.0 (1–4) | 1.0 (1–5) | 2.0 (1–5) | 2.0 (1–4) | 1.0 (1–4) | 56.6 (16.9) |
| p-value[b] | 0.036 | 0.085 | 0.663 | 0.415 | 0.236 | 0.955 |

(*Continued*)

**Table 3.** (Continued)

| Self-Rated Health Condition | | | | | | |
|---|---|---|---|---|---|---|
| Excellent | 1.0 (1–2) | 1.0 (1–1) | 1.0 (1–3) | 2.0 (1–2) | 1.0 (1–3) | 92.7 (6.8) |
| Good | 1.0 (1–3) | 1.0 (1–4) | 1.0 (1–5) | 1.0 (1–3) | 1.0 (1–3) | 73.2 (13.8) |
| Moderate | 2.0 (1–5) | 1.0 (1–4) | 2.0 (1–5) | 3.0 (1–5) | 2.0 (1–5) | 55.1 (17.6) |
| Poor | 2.5 (1–4) | 1.5 (1–5) | 3.0 (1–5) | 3.0 (1–4) | 2.0 (1–4) | 31.7 (16.1) |
| p-value[b] | <0.001 | 0.003 | <0.001 | <0.001 | 0.257 | <0.001 |
| History of Medical Condition(s) | | | | | | |
| Hypertensive | 1.0 (1–5) | 1.0 (1–5) | 1.0 (1–5) | 2.0 (1–4) | 1.0 (1–5) | 59.9 (21.7) |
| Non-Hypertensive | 2.0 (1–4) | 1.0 (1–4) | 2.0 (1–5) | 2.0 (1–5) | 2.0 (1–4) | 61.8 (29.9) |
| p-value[b] | 0.321 | 0.753 | 0.123 | 0.348 | 0.061 | 0.624 |

[a]Independent samples t-test/One-Way Analysis of Variance

[b]Mann-Whitney U test/Kruskall-Wallis test;

quality of life. People with poor health are likely to be those individuals with underlying medical conditions, which makes medicine necessary to improve their health. Therefore, this situation is more likely to elevate the level of burden that could be correlated with a negative impact on health-related quality of life. In the present study, the perceived burden of medication was not found to be significantly related to the number of medicines used and the number of underlying medical conditions.

This study revealed that the majority of the participants reported experiencing minimal (26.9%) to moderate (44.5%) degrees of perceived burden of medication. These findings differ from the estimates reported in Qatar [28]: minimal (66.8%) to moderate (24.1%); in England [29]: minimal (33.1%) to moderate (54.6%); and in Kuwait [30]: minimal (35.4%) to moderate (62.0%) degrees of burden respectively. In addition, the median perceived burden of medication captured by the present study was 100 (moderate degree of burden), which is higher than in both Qatar [28] (95) and England [31] (99.7), but lower than in Kuwait [30] (112). Furthermore, the median overall burden of treatment for the current study was reported to be 2 (minimal burden), which is lower than that found in the study conducted in England [31] (5 = some degree of burden), Qatar [28] (3 = minimal burden) and Kuwait [30] (5 = some degree of burden). The obvious differences between the various countries raise important questions about

**Table 4. Regression models showing factors associated with perceived burden of medication and perceived health-related quality of life.**

| Independent variables | Perceived Burden of Medication (LMQ-3 score) | | | Perceived Health-Related Quality of Life (EQ Vas Score) | | |
|---|---|---|---|---|---|---|
| | B (SE) | 95% CI | p-value | B (SE) | 95% CI | p-value |
| Age | -0.08 (0.1) | -0.31, 0.15 | 0.499 | 0.01 (0.1) | -0.22, 0.24 | 0.922 |
| No. of medicines | -2.64 (2.7) | -7.95, 2.67 | 0.327 | 1.02 (2.6) | -4.21, 6.26 | 0.699 |
| No. of medical conditions | 0.57 (1.45) | -2.32, 3.45 | 0.697 | -1.28 (1.28) | -3.81, 1.25 | 0.319 |
| Self-rated health condition | 8.84 (2.5) | 3.94, 13.74 | 0.001 | -18.77 (2.4) | -23.61, -13.94 | <0.001 |
| Overall burden of treatment (VAS-burden) | 3.37 (0.6) | 2.19, 4.56 | <0.001 | -1.01 (0.6) | -2.16, 0.14 | 0.086 |
| Hypertensive | -7.52 (3.6) | -14.60, -0.45 | 0.037 | | | |
| Model statistics | $R^2$ | Adjusted $R^2$ | p-value | $R^2$ | Adjusted $R^2$ | p-value |
| | 0.648 | 0.389 | <0.001 | 0.671 | 0.425 | <0.001 |

cultural attitudes to health and illness. This is an important area for further study in a multicultural community, like Australia, where it has important implications for medical practice.

Due to the significant positive correlation between age, number of medical conditions and number of medicines used, it was important to adjust for these associations in the subsequent regression analyses. Factors such as overall burden of treatment, self-rated health condition and being hypertensive were significant in predicting perceived burden of medication. In the present study, the majority of participants, who were aged 60 years and above, were reported to be hypertensive. Generally, hypertensive patients require as many as seven medicines to control their blood pressure. Specifically, hypertensive patients under 35 years of age require at least three drugs to achieve target blood pressure levels, while older hypertensive patients tend to require more drugs than younger patients [32]. It is possible that these older hypertensive participants need more medications to help them control their blood pressure, which could be related with a higher level of overall burden of treatment. All these factors were significantly predicting the degree of perceived burden of medication.

In terms of the predictors for health-related quality of life, only self-rated health condition and overall burden of treatment were found to be significant factors. Individuals with poor health are more likely to take more medicines to help improve their health, which could possibly be associated with higher overall burden of treatment, which was found to be associated with lower health-related quality of life.

## Strengths and limitations

The strengths of this study include (1) a significant sample dominated by participants aged 60 years and above; and (2) an emerging finding to suggest a possible correlation between underlying medical conditions and perceived burden of medication.

On the other hand, there were some limitations to this study. Firstly, the small sample size was lower than anticipated. Secondly, the responses captured through self-report questionnaires were likely to be associated with self-selection bias and recall bias. Thirdly, the bias due to non-participation was likely to affect the external validity of the study. Fourthly, all participants were recruited from one hospital due to logistic challenges. Therefore, it is important to replicate this study using a larger sample representing a range of different ethnic groups to further explore the findings of this study.

## Conclusions

This study has further supported existing literature by showing that increasing age is found to be correlated with lower degrees of perceived burden of medication, lower overall burden of treatment and lower health-related quality of life. A more comprehensive understanding of perceived burden of medication and overall burden of treatment using LMQ-3, as well as health-related quality of life using EQ-5D-5L provide opportunities for physicians to develop customised therapeutic care plans to achieve optimal clinical outcomes for their patients.

## Acknowledgments

The authors would like to thank all participants for their contributions to this study.

## Author Contributions

**Conceptualization:** Won Sun Chen, Md. Rafiqul Islam, Edward Ogden.

**Data curation:** Won Sun Chen, Md. Rafiqul Islam, Sajini Ambepitiya, William Sim, Wai Yiu, Joseph Carey, Edward Ogden.

**Formal analysis:** Won Sun Chen.

**Methodology:** Won Sun Chen, Md. Rafiqul Islam, Sajini Ambepitiya, William Sim, Wai Yiu, Joseph Carey, Edward Ogden.

**Project administration:** Won Sun Chen, Md. Rafiqul Islam, Sajini Ambepitiya, William Sim, Wai Yiu, Joseph Carey, Edward Ogden.

**Resources:** Won Sun Chen, Md. Rafiqul Islam, Sajini Ambepitiya, William Sim, Wai Yiu, Joseph Carey, Edward Ogden.

**Software:** Won Sun Chen.

**Supervision:** Won Sun Chen, Md. Rafiqul Islam, Edward Ogden.

**Validation:** Won Sun Chen.

**Writing – original draft:** Won Sun Chen, Md. Rafiqul Islam, Sajini Ambepitiya, William Sim, Wai Yiu, Joseph Carey, Edward Ogden.

**Writing – review & editing:** Won Sun Chen, Md. Rafiqul Islam, Sajini Ambepitiya, William Sim, Wai Yiu, Joseph Carey, Edward Ogden.

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
