## [Decision Letter · Decision Letter 0]

22 Feb 2022

PONE-D-21-36336Factors associated with patient experiences of the burden of using medicines and health-related quality of life: a cross-sectional studyPLOS ONE

Dear Dr. Chen,

Thank you for submitting your manuscript to PLOS ONE. After careful consideration, we feel that it has merit but does not fully meet PLOS ONE’s publication criteria as it currently stands. Therefore, we invite you to submit a revised version of the manuscript that addresses the points raised during the review process.

We look forward to receiving your revised manuscript.

Kind regards,

Sheikh Mohd Saleem, MBBS, MD

Academic Editor

PLOS ONE

Journal Requirements:

Additional Editor Comments (if provided):

Thank you for submitting your manuscript to PLOS One. We have arrived at a decision to consider this only after a major revision. The detained comments from the reviewers are attached. Kindly answer those queries in your revised manuscript and see if there are any issues which can be addressed and may have skipped at this moment.

Reviewers' comments:

Reviewer's Responses to Questions

**Comments to the Author**

1. Is the manuscript technically sound, and do the data support the conclusions?

Reviewer #1: Yes

Reviewer #2: Partly

2. Has the statistical analysis been performed appropriately and rigorously? 

Reviewer #1: Yes

Reviewer #2: Yes

3. Have the authors made all data underlying the findings in their manuscript fully available?

Reviewer #1: Yes

Reviewer #2: Yes

4. Is the manuscript presented in an intelligible fashion and written in standard English?

Reviewer #1: Yes

Reviewer #2: Yes

5. Review Comments to the Author

Reviewer #1: The manuscript is written in a meaningful manner systematically. Appropriate statistical analytical tests have been used for obtaining the objectives for which the study was planned for. The authors also have submitted the limitations of their study which are acceptable

On the whole the article is well written and is an evidence towards the outcome, which can be used for a broader study to be utilised for the public practice by the physicians.

Reviewer #2: 1. Sampling strategy should be mentioned in the methodology.

How the participants were invited?

what could be biased due to non-participation?

2. What is the data collection procedure? Is it self-administered?

3. The parameters taken for sample size estimation have not been mentioned. Even post-hoc power calculation could be an option.

4. Kindly check for tense and grammar once.

5. In some references issue number is not mentioned.

6. PLOS authors have the option to publish the peer review history of their article (what does this mean?). If published, this will include your full peer review and any attached files.

Reviewer #1: **Yes: **PRASANNA KAMATH B T

Reviewer #2: No

---

## [Author Response · Author response to Decision Letter 0]

15 Mar 2022

Dear Editor,

On behalf of my co-authors, I would like to thank you for providing valuable feedback to our manuscript entitled: ‘Factors associated with patient experiences of the burden of using medicines and health-related quality of life: a cross-sectional study’. 

After some discussions, the authors have decided to deposit our data to Dryad Digital Repository (doi:10.5061/dryad.d2547d84m). The data can be accessed through the following link: https://datadryad.org/stash/share/6Ubc3sxxDr40IqJyUPue5dYY_QTI4qrpFaFWeiPD51Q.

Please let me know if you have any further questions, thank you for your understanding! 

---

## [Decision Letter · Decision Letter 1]

12 Apr 2022

Factors associated with patient experiences of the burden of using medicines and health-related quality of life: a cross-sectional study

PONE-D-21-36336R1

Dear Dr. Chen,

We’re pleased to inform you that your manuscript has been judged scientifically suitable for publication and will be formally accepted for publication once it meets all outstanding technical requirements.

Kind regards,

Sheikh Mohd Saleem, MBBS, MD

Academic Editor

PLOS ONE

Additional Editor Comments (optional):

Reviewers' comments:

Reviewer's Responses to Questions

**Comments to the Author**

1. If the authors have adequately addressed your comments raised in a previous round of review and you feel that this manuscript is now acceptable for publication, you may indicate that here to bypass the “Comments to the Author” section, enter your conflict of interest statement in the “Confidential to Editor” section, and submit your "Accept" recommendation.

Reviewer #1: All comments have been addressed

Reviewer #3: All comments have been addressed

2. Is the manuscript technically sound, and do the data support the conclusions?

Reviewer #1: Yes

Reviewer #3: Yes

3. Has the statistical analysis been performed appropriately and rigorously? 

Reviewer #1: Yes

Reviewer #3: Yes

4. Have the authors made all data underlying the findings in their manuscript fully available?

Reviewer #1: Yes

Reviewer #3: Yes

5. Is the manuscript presented in an intelligible fashion and written in standard English?

Reviewer #1: Yes

Reviewer #3: Yes

6. Review Comments to the Author

Reviewer #1: The author has responded for the previous queries and given plausible explanation. The article may be accepted for publication.

Reviewer #3: It is a well investigated study. Results are tabulated in a logical order. Results will appear more comprehensible if regression equations are given at appropriate places for appropriate variables so that reader could easily estimate the magnitude of the outcome

7. PLOS authors have the option to publish the peer review history of their article (what does this mean?). If published, this will include your full peer review and any attached files.

Reviewer #1: **Yes: **PRASANNA KAMATH B T

Reviewer #3: No

---

## [Editor Report · Acceptance letter]

20 Apr 2022

PONE-D-21-36336R1 

Factors associated with patient experiences of the burden of using medicines and health-related quality of life: a cross-sectional study 

Dear Dr. Chen:

I'm pleased to inform you that your manuscript has been deemed suitable for publication in PLOS ONE. Congratulations! Your manuscript is now with our production department. 

Kind regards, 

on behalf of

Dr. Sheikh Mohd Saleem 

Academic Editor

PLOS ONE